# Counterfactual Intervention in Attention Multiple Instance Learning For Digital Pathology

**Imane Chraki**[1,2]                                    IMANE.CHRAKI@CENTRALESUPELEC.FR

**Pierre Marza**[1,2]                                    PIERRE.MARZA@CENTRALESUPELEC.FR

**Stergios Christodoulidis**[1,2]      STERGIOS.CHRISTODOULIDIS@CENTRALESUPELEC.FR

**Maria Vakalopoulou**[1,2]                MARIA.VAKALOPOULOU@CENTRALESUPELEC.FR

[1] *MICS Laboratory, CentraleSupélec, Université Paris-Saclay*

[2] *IHU PRISM, National Center for Precision Medicine in Oncology, Gustave Roussy*

**Editors:** Accepted for publication at MIDL 2026

## Abstract

Attention-based Multiple Instance Learning (MIL) has become a prominent framework for analysing whole-slide images (WSI). These models have been shown to achieve good performance on classification tasks, while also offering an inherent proxy for interpretability through attention weights. In this work, we first question the validity of using attention for the interpretability of MIL models. Subsequently, we propose Counterfactual Intervention in Attention for MIL (CIA-MIL), a causal extension of attention-based MIL that explicitly measures and optimizes the contribution of attention to slide-level predictions. Across four histopathology classification benchmarks (BRCA, NSCLC, LUAD, Camelyon16) and two feature encoders (Resnet50, UNI), we investigate how the interpretability of attention relates to the representation space, and the downstream performance. We then show that CIA-MIL achieves performance comparable to strong MIL baselines while providing a more causally meaningful attention vector for explaining the model's outcome. Qualitative perturbation experiments show that dropping the top-attended patches leads to a larger confidence degradation in CIA-MIL compared to baseline ABMIL, highlighting the potential of causal supervision for reliable and interpretable WSI-based prediction.

**Keywords:** Multiple Instance Learning, Attention, Interpretability, Whole Slide Images, Digital Pathology, Causal Intervention.

## 1. Introduction

Multiple Instance Learning (MIL) is a weakly supervised learning framework designed for scenarios in which data are organized into sets of instances, referred to as bags, while supervision labels are only provided at the bag level (Dietterich et al., 1997; Maron and Lozano-Pérez, 1997). MIL is a relevant framework in many domains where fine-grained annotations are scarce or expensive, such as digital histopathology (Amores, 2013; Wang et al., 2016; Diao et al., 2021; Campanella et al., 2019), where labels are typically provided at the level of Whole Slide Images (WSIs), which can reach extremely large resolutions (often as large as 100k pixels per side)(Lu et al., 2021b).

In the standard MIL assumption, a bag is labelled positive if at least one of its instances is positive, and negative otherwise. While this assumption is appropriate in certain detection tasks, it is often restrictive for real-world biomedical applications. In practice, slide-level

labels usually arise from complex interactions between multiple tissue regions rather than from a single discriminative patch. To address this limitation, numerous attention-based MIL approaches have been proposed, enabling models to learn instance-level importance directly from data (Ilse et al., 2018; Shao et al., 2021; Li et al., 2021). These methods replace strategies such as max or mean pooling with learnable attention mechanisms that assign relevance scores to each instance, referred to as attention.

Despite the remarkable success of attention-based MIL in digital pathology, important questions remain regarding the reliability of attention as an interpretability tool. Attention weights are often visualized as heatmaps and interpreted as indicators of model reasoning (Lu et al., 2021a; Wagner et al., 2023). However, recent work on interpretability has shown that attention may not reliably reflect the true importance of instances for the final prediction (Hense et al., 2024; Zhang et al., 2022; Early et al., 2023; Javed et al., 2022). In some cases, models can achieve similar predictions while relying on substantially different attention distributions, revealing a potential disparity between attention and decision-making. This limitation is particularly critical in medical applications, where interpretability is not merely a convenience but a prerequisite for trust, clinical adoption, and regulatory approval.

In this work, we investigate the reliability of raw attention scores as a proxy for interpretability in MIL models. We study a broad family of attention-based MIL architectures within a unified experimental framework and quantitatively and qualitatively report how different design choices such as attention type, multihead formulations, clustering strategies, and entropy regularization affect the attention effect on predictions. In a second part, we explore the use of causal counterfactual intervention to guide the learning of attention toward representations that are more causally aligned with the model's predictions. By enforcing counterfactual consistency during training, our goal is to promote attention patterns that reflect more accurately the underlying causal mechanisms. This introduces an explicit trade-off between predictive performance and interpretability, which we characterize empirically.

The main contributions of this work can be summarized as follows: *(i) Attention reliability analysis*, we perform an extensive evaluation of attention reliability across a wide range of MIL models, including standard attention MIL and its variants such as clustering-based MIL, multi-head attention MIL, and DSMIL. Our analysis reveals that the attention byproduct of current state-of-the-art MILs does not fully align causally with downstream prediction. *(ii) Novel counterfactual attention intervention framework (CIA-MIL)*, we propose a novel counterfactual-guided attention learning strategy designed to improve the causal alignment and stability of attention mechanisms in MIL models. *(iii) Analysis of the interplay between downstream performance and attention interpretability*, we conduct extensive experiments on real-world digital pathology datasets, demonstrating the complex trade-offs between predictive performance and interpretability of attention. Importantly, we do not claim to provide a method to increase downstream performance, but rather to maintain the performance of current MIL methods while improving their explainability, as this is crucial when considering the safe deployment of medical imaging techniques.

## 2. Related Work

**Attention-Based MIL** — Embedding-level MIL models operate directly in the instance embedding space to compute a compact bag-level representation that is subsequently passed to a classifier. In the standard attention-based MIL formulation (Ilse et al., 2018), the bag representation is expressed as a weighted sum of instance embeddings, where the attention weights are learned functions of the instances themselves. Building upon this framework, numerous variants have been proposed to improve representational capacity and performance. TransMIL (Shao et al., 2021) introduces self-attention (Wagner et al., 2023; Xiong et al., 2021), while CLAM (Lu et al., 2021b) integrates clustering-based attention to capture multiple discriminative regions within a bag. DSMIL (Li et al., 2021) adopts a dual-stream architecture to explicitly model instance-level and bag-level interactions. Other extensions leverage multi-head attention and multi-branch aggregation strategies. More recent models aim to regularize attention, reducing over-reliance on a few highly activated instances. For instance, MHIM-MIL (Tang et al., 2023) adopts a Siamese framework with masked attention to mine hard-to-classify instances, while ACMIL (Zhang et al., 2024) introduces multi-branch attention together with stochastic top-$K$ instance masking to promote diversity in discriminative patterns. To further improve learning under limited supervision, data distillation and pseudo-bag generation strategies such as DTFD-MIL (Zhang et al., 2022) have also been proposed.

**Interpretability of MIL Models** — In digital histopathology, attention-based MIL models typically rely on attention scores to generate patch-level relevance maps that highlight regions of interest. Despite their intuitive appeal, several studies have shown that raw attention maps do not necessarily provide faithful explanations of model behavior (Hense et al., 2024; Javed et al., 2022; Zhang et al., 2022). To address this limitation, alternative explainability strategies have been proposed. DTFD (Zhang et al., 2022) reframes MIL as an equivalent image classification problem and derives patch-level importance scores using Grad-CAM (Selvaraju et al., 2017). Other post-hoc explainability strategies have also been proposed (Adebayo et al., 2018; Kindermans et al., 2019; Pirovano et al., 2020; Wang et al., 2019; Bach et al., 2015; Baehrens et al., 2010; Shrikumar et al., 2017; Montavon et al., 2019; Hense et al., 2024), including perturbation-based approaches (Early et al., 2023), while fully additive models explicitly decompose the bag-level prediction into a sum of instance contributions (Javed et al., 2022). Despite these advances, many existing interpretability methods suffer from high computational cost, limited scalability to large bags, or simplifying assumptions that neglect complex inter-instance dependencies. Developing MIL models that are simultaneously accurate, scalable, and faithfully interpretable remains an open challenge.

**Causal Inference in MIL** — Recently, causality (Pearl and Mackenzie, 2018; Pearl et al., 2016) has been introduced into the MIL framework to account for confounding factors that may compromise model training. Models such as CAMIL (Chen et al., 2024a) and CATTMIL (Wu et al., 2024) formulate the bag-level representation as a mediator between patch embeddings and the final prediction by applying a front-door adjustment (Pearl et al., 2016). In contrast, IBMIL (Lin et al., 2023) adopts a back-door adjustment (Pearl et al., 2016) strategy via a two-stage training procedure to explicitly control for co-founders. These approaches highlight the growing interest in causal reasoning within MIL pipelines, particularly for medical imaging applications, where biased signals affecting images might degrade

prediction reliability. In our work, instead of reasoning in the image space, we operate directly at the attention level. We climb the causality ladder higher than the adjustment level and apply counterfactual intervention in attention. The objective is not bias removal, but improving the faithfulness of inherent attention to directly explain the downstream prediction.

**Counterfactual Intervention for Attention Learning** — Counterfactual analysis provides a principled framework to measure the causal influence of input features on model predictions. In computer vision, counterfactual attention learning has been introduced to guide attention mechanisms through causal supervision rather than relying solely on conventional likelihood maximization. Notably, (Rao et al., 2021) proposes a counterfactual attention learning framework that explicitly maximizes the prediction difference between factual and counterfactual attentions to encourage the discovery of causally effective visual regions. While these ideas have demonstrated strong performance in fine-grained recognition and re-identification tasks, their integration into multiple instance learning for digital pathology remains largely unexplored. Our work bridges this gap by formulating counterfactual causal supervision directly at the MIL aggregation level.

## 3. Methods

### 3.1. Attention-Based Multiple Instance Learning

We adopt the attention-based MIL formulation of (Ilse et al., 2018). A bag $X = \{x_i\}_{i=1}^{N}$ consists of $N$ instances, each mapped to a feature embedding by a shared frozen encoder:

$$\mathbf{z}_i = f(x_i), \qquad i = 1, \ldots, N. \tag{1}$$

The instance embeddings are aggregated into a bag-level representation using a learnable attention-weighted pooling, $\mathbf{Z} = \sum_{i=1}^{N} a_i \mathbf{z}_i$ which is passed to a classifier to obtain the final bag prediction $Y = \varphi(\mathbf{Z})$.

### 3.2. Counterfactual Attention Intervention

To explicitly model the causal contribution of attention to prediction, we model an attention-based MIL framework through a structural causal model (SCM) graph (Pearl et al., 2016) as shown in Fig.1, where $X$: WSI (bag), $Y$: bag label, $A$: attention distribution. $X \to A$ indicates that attention is generated from $X$, $A \to Y$ indicates that attention leads to a bag level prediction, and $X \to Y$ indicates that bag instances lead to a bag level prediction. Let $X = \{z_i\}_{i=1}^{N}$ denote the instance embeddings and $A = \{a_i\}_{i=1}^{N}$ the learned attention distribution. The standard prediction is:

$$Y(A, X) = \varphi\left(\sum_{i=1}^{N} a_i z_i\right) = \varphi(\mathbf{Z}) \tag{2}$$

We introduce a counterfactual intervention in attention by cutting the path from $X$ to $A$ and measuring the effect of this intervention on the prediction as shown in Fig.1(b).

$$Y(\mathrm{do}(A = \bar{A}), X) = \varphi\left(\sum_{i=1}^{N} \bar{a}_i z_i\right) = \bar{Y} \tag{3}$$

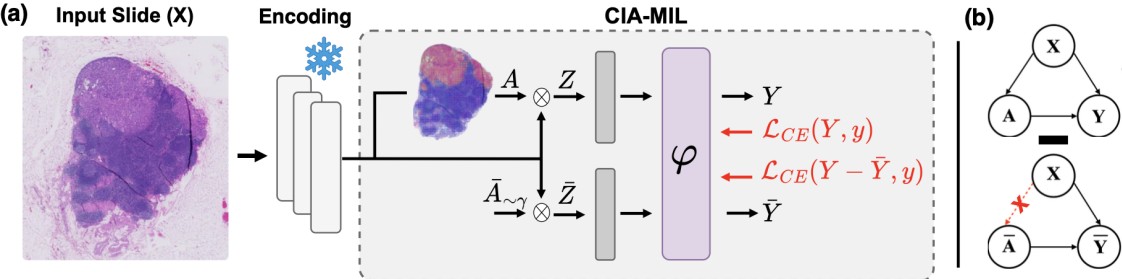

Figure 1: **Overview of CIA-MIL and its causal intervention module.** (a) CIA-MIL: Proposed model architecture. (b) Structural causal graph of the counterfactual attention intervention.

where the do-operation $\text{do}(\cdot)$ forcibly assigns a specific non-informative value to $A$, referred to as $\bar{A}$, while keeping $X$ fixed. $\bar{A} \sim \gamma$ may be sampled as random or uniform attention. This way, the attention effect is defined as:

$$Y_{\text{effect}} = \mathbb{E}_{\bar{A}\sim\gamma}\big[Y(A, X) - Y(\text{do}(A = \bar{A}), X)\big] \tag{4}$$

In practice, according to the results in (Baldi and Sadowski, 2014), the expectation is approximated by sampling a single counterfactual attention per bag, resulting in negligible training overhead and no additional inference cost. To guide the learning of causally effective attention, we add a cross-entropy loss between the effect caused by the counterfactual intervention and the ground-truth bag label $y$. The final training objective is:

$$\mathcal{L} = \mathcal{L}_{\text{CE}}(Y(A, X), y) + \lambda \, \mathcal{L}_{\text{CE}}(Y_{\text{effect}}, y) \tag{5}$$

where $\lambda$ is a hyperparameter, and it controls the influence of the counterfactual supervision. The additional causal intervention loss term to the standard classification cross-entropy loss guides the model to not only learn the classification task, but also to ensure attention patterns that are effect causing for the model: a well-learned attention $A$ should lead to a correct prediction, while a random non-informative attention $\bar{A}$ should fail in leading to the same level of prediction accuracy.

### 3.3. Attention-Based Perturbation Analysis

We assess the faithfulness of MIL attention as an interpretability proxy using a region perturbation strategy (Hense et al., 2024; Alber et al., 2019). Patches are ranked by decreasing attention scores and partitioned into 100 disjoint subsets $\{r_1, \ldots, r_{100}\}$ of equal size. The perturbed slide at step $k < 100$ is defined as:

$$X^{(k)} = \bigcup_{i=k+1}^{100} r_i, \qquad k = 0, \ldots, 99, \tag{6}$$

The model's prediction is evaluated at each perturbation step to obtain a perturbation curve. For every slide $X$, we start first with prediction from the original bag with all

patches $X^{(0)} = X$, and start removing subsequently at each following step $k$, subset $r_k$, to get the model's prediction $s(k)$ for the perturbed slide $X^{(k)}$. To evaluate the attention influence on prediction, we calculate the Area Under the Perturbation Curve (AUPC). A lower AUPC indicates a faster degradation of predictive confidence when highly attended regions are removed, reflecting an attention mechanism valid for explaining the model's decision. Additional details are provided in Appendix C.

## 4. Experimental Setup

**Datasets** — We conduct our experiments on whole-slide images (WSIs) from multiple publicly available datasets from TCGA (Tomczak et al., 2015): TCGA-NSCLC (Non–Small Cell Lung Cancer) and TCGA-BRCA (Breast Invasive Carcinoma) for the binary subtyping task, TCGA-LUAD (Lung Adenocarcinoma) for TP53 mutation prediction, Camelyon16 (Bejnordi et al., 2017) for binary metastasis detection in breast lymph node.

**Implementation Details** — Slides were processed using the same approach as in (Lu et al., 2021b) to obtain patches of size 256x256 at 20x magnification. We use pre-extracted features with two different image encoders: **(i)** ResNet50 (He et al., 2016) pre-trained on ImageNet (Deng et al., 2009) and **(ii)** UNI-V1 (Chen et al., 2024b) foundation model, to test performance in both in and out of domain pre-training. Please note that any other encoder could be integrated into our method. Experiments were conducted under five cross-validation settings for TCGA-NSCLC, TCGA-BRCA, and TCGA-LUAD using a learning rate of 0.0002. We used the originally published train-test split for Camelyon16 in three runs with a learning rate of 0.0002. All experiments were conducted using the Adam optimizer (Kingma, 2014) with a maximum total of 150 epochs and an early stopping criterion of 50 epochs.

## 5. Results and Discussion

We evaluate and challenge our method against standard ABMIL (Ilse et al., 2018) and its variants, where additional modules are added to mitigate problems encountered during training relative to attention, through clustering (CLAM-SB (Lu et al., 2021b)), multi-head attention and attention masking (ACMIL (Zhang et al., 2024)), attention stage (AddMIL (Javed et al., 2022)), hard instances mining in a 2-stage training framework (MHIM (Tang et al., 2023)). As well as DSMIL (Li et al., 2021), where attention is calculated in a dual stream manner, but the model still satisfies equations 1 and 2. We compare downstream performance also against instance-based baselines, namely average and maximum pooling strategies (MeanMIL/MaxMIL) that do not contain attention modules. We also compare against IBMIL (Lin et al., 2023) on Camelyon16. When needed, and if not explained differently, raw attention of models is used to assess their reliability. We report downstream tasks performance metrics in terms of Area Under the Curve (AUC) and F1 score, and AUPC on correct predictions to assess attention as an explainability method. For Camelyon 16, for which we have access to fine-grained annotations at the level of patches, we report the AUPRC, the area under the precision-recall curve for attention as a prediction of instance labels, using a sigmoid operator. We further assess AOPCR consistent with prior work (Early et al., 2023), which contrasts targeted perturbation with random perturbation to

assess the informativeness of attention rankings in comparison with a random ranking of patches, and pointing game of top 5 patches PG@5, defined as the percentage of metastatic test slides for which at least one of the top-5 most attended patches overlaps an annotated metastasis region.

**TCGA Benchmark** — We evaluate all models on BRCA and NSCLC tumor subtyping and on the more challenging LUAD TP53 mutation prediction task (Table 1). Across BRCA and NSCLC, most attention-based MIL models, particularly ABMIL-derived methods, achieve very high AUC, often exceeding 0.95 with in-domain foundation models such as the UNI features. However, this strong predictive performance is accompanied by large variability in attention faithfulness. ABMIL-based models frequently exhibit elevated AUPC across slides as reported by the elevated standard deviations, whereas DSMIL shows a more favorable performance-explainability trade-off, although with sensitivity to the feature encoder. CIA-MIL applied with baseline ABMIL achieves among the lowest AUPC while maintaining competitive AUC across both feature extractors, reflecting an explicit and stable performance–explainability balance. On LUAD, where overall performance drops across all methods, CIA-MIL still achieves low AUPC with competitive AUC, indicating that counterfactual supervision remains effective even in low-signal data. An important insight is that when attention-based MIL models already achieve strong downstream performance, introducing the causal intervention primarily serves to guide attention toward more causally meaningful patterns. Conversely, in settings where predictive performance is weaker, the intervention can also stabilize and improve the training process itself, leading to more reliable attention and better overall performance trade-off.

Table 1: **TCGA Benchmark: Performance vs Attention Interpretability.** Comparison of classification performance (AUC ↑) and attention faithfulness (AUPC ↓) across three tasks (BRCA subtyping, NSCLC subtyping, LUAD TP53 mutation prediction) and two feature extractors (ResNet50, UNI).

| | BRCA - ResNet50 | | BRCA - UNI | | NSCLC - ResNet50 | | NSCLC - UNI | | LUAD - ResNet50 | | LUAD - UNI | | Average | |
| | AUC (↑) | AUPC (↓) | AUC (↑) | AUPC (↓) | AUC (↑) | AUPC (↓) | AUC (↑) | AUPC (↓) | AUC (↑) | AUPC (↓) | AUC (↑) | AUPC (↓) | AUC (↑) | AUPC (↓) |
|---|---|---|---|---|---|---|---|---|---|---|---|---|---|---|
| MeanMIL | 89.0±3.8 | N/A | 93.2±2.4 | N/A | 91.1±3.0 | N/A | 96.9±1.3 | N/A | 66.9±6.5 | N/A | 74.5±6.3 | N/A | 85.3 | N/A |
| MaxMIL | 86.9±2.6 | N/A | **95.4±1.5** | N/A | 94.4±1.6 | N/A | 97.5±1.0 | N/A | 61.4±9.2 | N/A | 76.0±5.4 | N/A | 85.3 | N/A |
| DSMIL | 88.4±3.0 | 84.0±17.3 | 94.1±1.6 | **35.1±26.8** | 93.6±2.5 | 65.4±32.4 | 97.4±1.1 | **35.6±28.6** | 67.8±6.3 | 80.4±18.8 | 66.9±4.7 | **56.8±29.0** | 84.7 | 59.5 |
| AddMIL | 88.0±2.0 | 72.4±28.0 | 93.7±2.6 | 71.7±33.8 | 92.0±2.3 | **56.5±48.8** | 94.6±3.0 | 62.6±41.6 | 63.4±3.8 | 85.3±27.8 | 73.3±3.9 | 81.9±25.3 | 84.2 | 71.7 |
| MHIM | 90.0±1.9 | 86.4±20.9 | 94.2±1.1 | 73.5±23.9 | 94.8±1.5 | 67.9±33.0 | 97.9±0.9 | 77.9±26.0 | 68.0±3.1 | 78.8±20.2 | 73.8±3.7 | 77.3±26.9 | 86.4 | 76.9 |
| ABMIL | 89.2±2.6 | 85.5±22.4 | 95.3±1.7 | 76.1±25.3 | 93.4±1.7 | 65.9±42.7 | 97.6±1.0 | 70.7±33.6 | 68.2±5.6 | 78.0±24.8 | 74.0±4.3 | 80.9±26.2 | 86.3 | 76.2 |
| CIA-MIL | 90.1±2.8 | 68.8±16.5 | 94.5±1.2 | 71.2±21.8 | 91.6±2.4 | 62.8±23.2 | 96.5±1.6 | 64.0±23.7 | 69.1±5.1 | 73.3±20.4 | 77.5±2.5 | 73.1±24.0 | 86.6 | 68.8 |
| CLAM | 89.4±1.8 | 84.9±21.9 | 94.5±2.3 | 73.1±25.5 | 94.1±1.9 | 64.6±44.3 | 97.9±0.8 | 74.1±29.0 | 65.5±6.2 | 76.5±34.5 | 74.0±3.0 | 78.5±26.1 | 85.9 | 75.3 |
| CIA-CLAM | 85.9±6.3 | 62.5±23.8 | 94.9±1.7 | 72.8±21.7 | 92.0±3.2 | 63.7±18.0 | 97.8±1.0 | 72.2±23.3 | 67.1±5.3 | 74.9±15.6 | 75.3±4.4 | 73.9±22.9 | 85.5 | 70.0 |
| ACMIL | 88.6±2.8 | 69.2±23.4 | 94.4±2.9 | 82.8±22.0 | 93.8±1.8 | 62.5±46.2 | **97.9±0.8** | 73.8±27.7 | 67.5±5.0 | 85.3±27.8 | 75.5±7.4 | 81.9±25.3 | 86.3 | 75.9 |
| CIA-ACMIL | **90.3±1.4** | 69.5±26.7 | 94.9±2.3 | 81.1±19.6 | 92.0±3.2 | 61.7±18.0 | 97.3±1.8 | 68.8±26.9 | 67.0±2.8 | **71.6±21.9** | 75.7±6.9 | 75.8±24.9 | 86.2 | 71.4 |

To further analyze the sensitivity of each model to attention-guided perturbations, Fig.2 presents the perturbation curves on BRCA and NSCLC. Models with causally faithful attention exhibit more pronounced degradation of prediction confidence when highly attended patches are removed, whereas models with diffuse attention remain comparatively insensitive. CIA-MIL consistently shows steeper decay profiles with respect to baseline ABMIL counterpart, confirming that counterfactual supervision effectively reshapes attention toward causally meaningful tissue regions for the model's prediction. We note, however, that limited prediction degradation under perturbation does not necessarily imply poor representations: attention mechanisms may still support strong feature learning and accurate bag-level predictions even when they are weakly causally coupled to the final decision. In

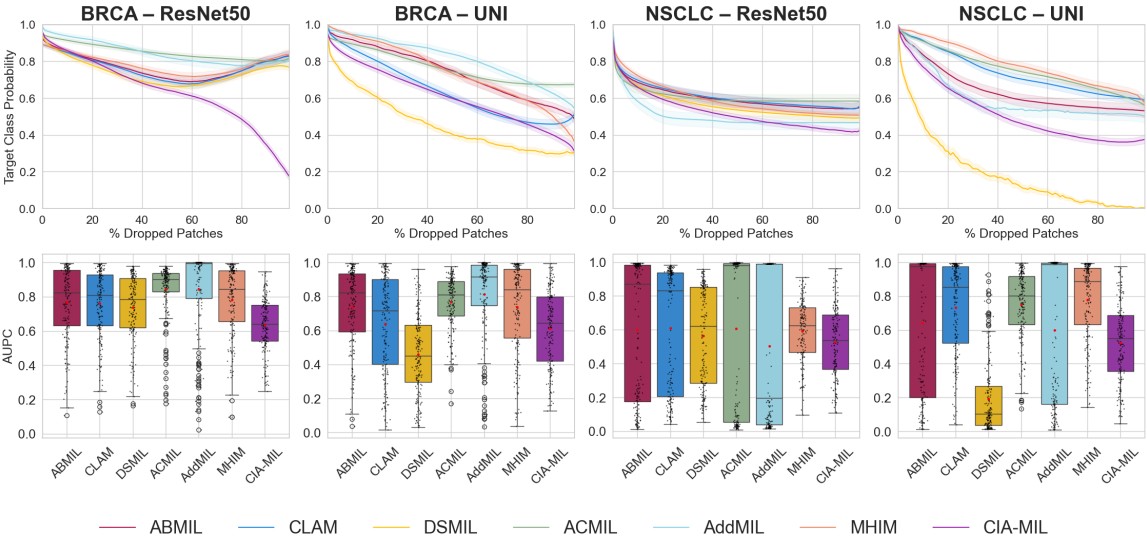

Figure 2: Attention-based perturbation analysis on the BRCA and NSCLC subtyping tasks. The curves show the evolution of the target class probability as increasingly larger fractions of highly attended patches are removed. Boxplots report statistics over slides.

such cases, attention should be interpreted as a latent aggregation mechanism rather than as a faithful explanation of the model's prediction.

**Camelyon16 Benchmark** — We evaluate models on Camelyon16 with UNI features to explicitly disentangle bag-level performance from instance-level interpretability. Table 2 reports bag-level classification performance (AUC, F1) together with instance-level attention faithfulness (AUPC, AOPCR) and localization accuracy (AUPRC, PG@5). AUPRC is computed by applying a sigmoid to raw patch attentions (Sig). For DSMIL, an additional normalization of raw attention with respect to the embedding dimensionality is applied to ensure comparability with ABMIL-based variants.

At the bag level, nearly all attention-based MIL models achieve near-saturated performance, with AUC values consistently above 98%. However, clear differences emerge at the instance level.

In particular, introducing CIA module achieves a favorable balance across instance-level metrics, combining lower AUPC and higher AOPCR with competitive AUPRC and localization performance. Importantly, this reduction in AUPC is accompanied by consistent agreement with annotated metastatic regions, as evidenced by AUPRC and PG@5, rather than by degenerate sparsity or loss of localization. This suggests that counterfactual intervention in attention yields attention that is simultaneously globally consistent, spatially selective, and causally aligned with outcome without compromising slide-level performance. We note, however, that when CIA is combined with methods that already include strong attention-level regularization mechanisms, such as CLAM, careful tuning of the intervention strength might be required. In such cases, an overly strong causal regularization may

interfere with existing attention constraints and lead to slight degradation in predictive performance.

Table 2: **Camelyon16: Bag-Level Performance vs Instance-Level Interpretability.** Comparison of bag-level classification performance (AUC, F1) and instance-level interpretability measured by AUPC and AUPRC. AUPRC (Sig) is computed by applying a sigmoid to raw patch-level attentions.

|  | Bag | | Explainability | | Interpretability | |
|---|---|---|---|---|---|---|
|  | **AUC** ($\uparrow$) | **F1** ($\uparrow$) | **AUPC** ($\downarrow$) | **AOPCR** ($\uparrow$) | **AUPRC**$_{sig}$ ($\uparrow$) | **PG@5** ($\uparrow$) |
| Meanmil | $62.5 \pm 4.8$ | $46.7 \pm 9.6$ | N/A | N/A | N/A | N/A |
| MaxMIL | $98.3 \pm 0.4$ | $94.2 \pm 1.9$ | N/A | N/A | N/A | N/A |
| DSMIL | $98.9 \pm 1.1$ | $97.9 \pm 1.1$ | $66.8 \pm 44.8$ | $30.0 \pm 40.2$ | $78.4 \pm 9.7$ | $20.8 \pm 8.8$ |
| AddMIL | $98.2 \pm 1.4$ | $95.0 \pm 3.4$ | $67.5 \pm 45.7$ | $28.4 \pm 40.4$ | $93.4 \pm 1.2$ | $87.5 \pm 0.0$ |
| MHIM | $98.2 \pm 1.1$ | $94.2 \pm 2.8$ | $72.9 \pm 51.2$ | $27.8 \pm 45.5$ | $93.2 \pm 0.5$ | $86.1 \pm 2.0$ |
| ABMIL | $98.7 \pm 0.3$ | $95.5 \pm 2.2$ | $69.6 \pm 43.8$ | $25.3 \pm 39.3$ | $93.2 \pm 1.2$ | $86.8 \pm 1.0$ |
| ABMIL-IBMIL | $\mathbf{99.9 \pm 0.2}$ | $\mathbf{99.3 \pm 1.2}$ | $68.2 \pm 42.7$ | $28.1 \pm 38.9$ | $91.5 \pm 2.4$ | $86.8 \pm 1.0$ |
| CIA-MIL | $99.2 \pm 0.5$ | $96.3 \pm 2.0$ | $\underline{56.8 \pm 23.8}$ | $\underline{41.0 \pm 22.7}$ | $92.9 \pm 1.5$ | $87.5 \pm 0.0$ |
| CLAM | $\underline{99.7 \pm 0.3}$ | $97.6 \pm 2.1$ | $67.8 \pm 44.4$ | $27.9 \pm 40.7$ | $94.4 \pm 0.3$ | $87.5 \pm 0.0$ |
| CIA-CLAM | $96.5 \pm 6.1$ | $93.0 \pm 11.3$ | $\mathbf{54.4 \pm 26.2}$ | $\mathbf{42.7 \pm 28.9}$ | $95.2 \pm 0.2$ | $86.8 \pm 1.0$ |
| ACMIL | $\underline{99.4 \pm 0.5}$ | $96.9 \pm 2.7$ | $68.2 \pm 45.3$ | $28.1 \pm 40.9$ | $\underline{95.4 \pm 0.5}$ | $87.5 \pm 0.0$ |
| CIA-ACMIL | $\mathbf{99.9 \pm 0.1}$ | $98.6 \pm 1.2$ | $60.8 \pm 32.0$ | $35.6 \pm 29.5$ | $\mathbf{95.5 \pm 0.4}$ | $87.5 \pm 0.0$ |

**Qualitative Attention Analysis.** Fig.3 provides a qualitative comparison of attention maps on representative crops from a metastatic Camelyon16 slide. The first row shows a metastatic region (red annotations), followed by the corresponding ABMIL, ACMIL and CIA-MIL attention overlays, while the second row displays a non-metastatic region. ABMIL exhibits more diffuse attention and assigns non-negligible attention to non-metastatic tissue. In contrast, ACMIL and CIA-MIL concentrate attention sharply within metastatic regions and suppress responses in non-metastatic areas.

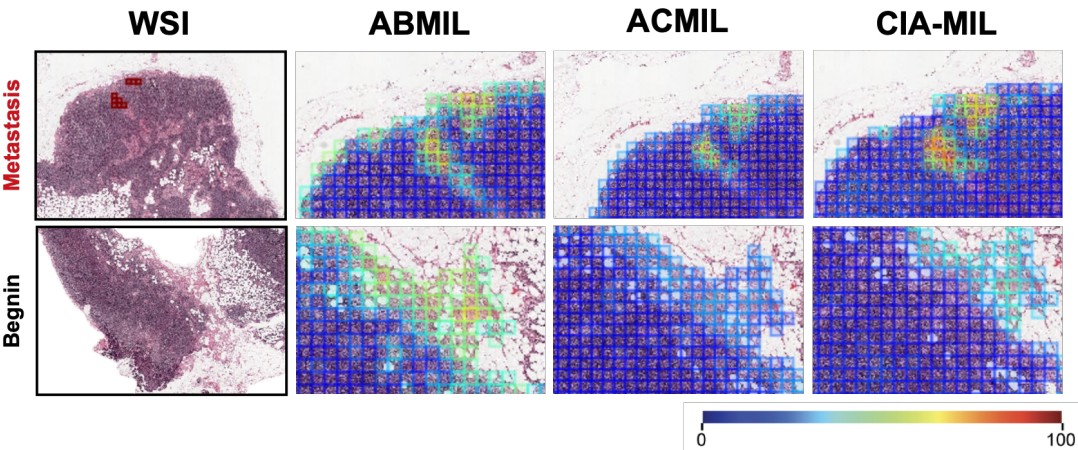

Figure 3: **Qualitative comparison of attention maps on a Camelyon16 WSI.** Attention is normalised with min-max slide within the slide for compared models: ABMIL, ACMIL, and CIA-MIL.

**Why Still Use ResNet in the Era of Foundation Models?** — Although foundation models such as UNI (Chen et al., 2024b) provide in-domain representations, we include ResNet50 (He et al., 2016) pretrained on ImageNet (Deng et al., 2009) as a feature extractor. In fact, evaluating both out-of-domain and in-domain features allows us to disentangle the effect of representation learning from that of attention supervision. Also, the fact that CIA-MIL exhibits consistent faithfulness improvements across both ResNet50 and UNI demonstrates that the proposed counterfactual intervention is not tied to a specific representation space. Notably, while UNI generally yields higher absolute AUC values, attention faithfulness as measured by AUPC does not automatically improve with stronger foundation features. In some cases, certain configurations of MIL models can achieve the same performance or even slightly degraded with foundation models compared to generic representations, as in the example of DSMIL on the LUAD TP53 mutation prediction as reported in Table 1. This further supports our central claim: improvements in representation power alone do not guarantee causally meaningful attention, and explicit counterfactual supervision of attention can help mitigate this issue.

## 6. Ablation Studies

**Random vs Uniform Counterfactual Intervention** — We study the impact of the counterfactual attention distribution used for intervention. Table 3 compares random and uniform counterfactual attentions across BRCA and NSCLC. Both strategies lead to consistent improvements in attention faithfulness compared to ABMIL, validating that the proposed causal mechanism is not sensitive to a specific choice of counterfactual distribution. However, subtle differences emerge between the two variants. Random counterfactual attention generally yields slightly lower AUPC, especially on NSCLC with UNI features, indicating stronger selective pressure on attention. Uniform counterfactual attention, by contrast, occasionally preserves marginally higher AUC at the cost of slightly degraded AUPC. This suggests that random intervention introduces stronger stochastic perturbations that better suppress spurious correlations, whereas uniform intervention acts as a weaker regularizer.

Table 3: **Intervention Choice: Random vs Uniform Counterfactual Attention Distribution.** Comparison of CIA-MIL using random and uniform counterfactual attentions across BRCA and NSCLC datasets with both ResNet50 and UNI feature extractors.

| | BRCA - ResNet50 | | BRCA - UNI | | NSCLC - ResNet50 | | NSCLC - UNI | |
|---|---|---|---|---|---|---|---|---|
| | AUC ($\uparrow$) | AUPC ($\downarrow$) | AUC ($\uparrow$) | AUPC ($\downarrow$) | AUC ($\uparrow$) | AUPC ($\downarrow$) | AUC ($\uparrow$) | AUPC ($\downarrow$) |
| ABMIL (Ilse et al., 2018) | 89.2±2.6 | 83.6±22.8 | **95.3±1.7** | 75.3±25.3 | **93.4±1.7** | 62.4±42.9 | **97.6±1.0** | 64.7±40.6 |
| CIA-MIL (Random) | 90.1±2.8 | 65.7±15.1 | 94.5±1.2 | 63.6±23.5 | 91.6±2.4 | 56.1±22.0 | 96.5±1.6 | 52.0±23.4 |
| CIA-MIL (Uniform) | **90.2±3.1** | 63.8±16.6 | 94.6±2.0 | 71.2±21.6 | 91.5±2.4 | 57.1±21.4 | 97.2±1.9 | 76.1±23.6 |

**Causal Effect Weighting: Performance-Interpretability Trade-off** — We analyse in Fig.4 the influence of the causal effect weighting coefficient $\lambda$ on the balance between predictive performance and attention interpretability. When $\lambda = 0$, the model reduces to a standard attention-based MIL baseline without causal supervision, resulting in weaker perturbation sensitivity. As $\lambda$ increases, the drop in AUPC increases on both BRCA and

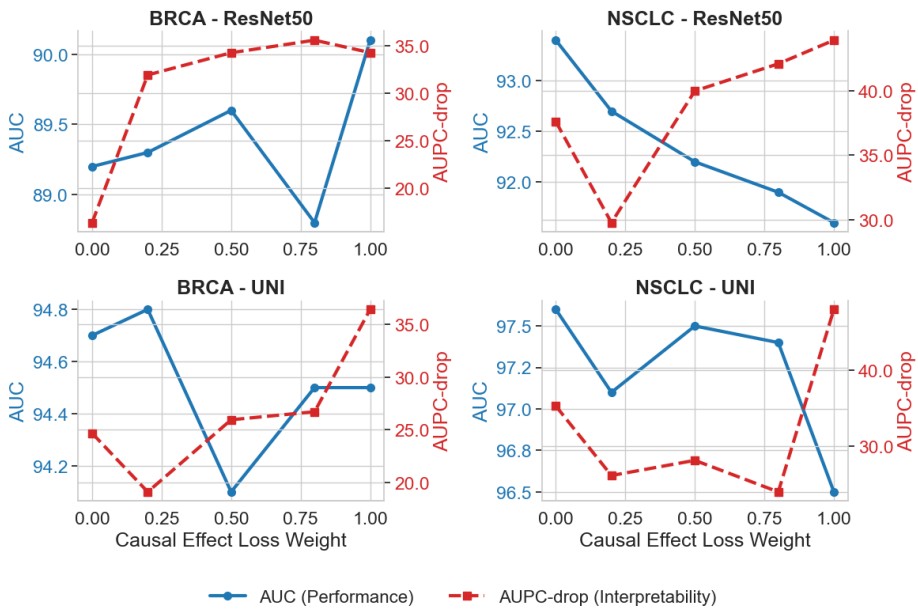

Figure 4: Performance-interpretability trade-off controlled by the causal effect weighting coefficient $\lambda$.

Table 4: Attention stability across folds measured by slide-wise Spearman correlation (%, mean $\pm$ std). We report (i) cross-architecture sensitivity by comparing ABMIL attentions to other MIL models, and (ii) within-architecture stability between baseline and CIA-augmented attentions .

|  | All Slides | Positive Slides | Positive Patches |
|---|---|---|---|
| ABMIL vs ACMIL | 71.4 ± 06.9 | 75.0 ± 04.9 | 76.4 ± 05.6 |
| ABMIL vs AddMIL | 84.3 ± 12.1 | 87.2 ± 09.3 | 86.8 ± 05.2 |
| ABMIL vs CLAM | 67.7 ± 15.5 | 72.8 ± 12.2 | 69.7 ± 02.8 |
| ABMIL vs DSMIL | 43.1 ± 08.3 | 58.2 ± 13.0 | 70.2 ± 04.4 |
| ABMIL vs IBMIL | 91.1 ± 02.9 | 92.6 ± 02.4 | 91.2 ± 02.1 |
| ABMIL vs MHIM | 57.1 ± 13.3 | 63.2 ± 12.4 | 65.1 ± 05.7 |
| ABMIL vs CIA-MIL | 34.6 ± 24.6 | 48.7 ± 19.3 | 79.8 ± 01.2 |
| ACMIL vs CIA-ACMIL | 76.8 ± 09.0 | 79.3 ± 06.6 | 84.9 ± 04.3 |
| CLAM vs CIA-CLAM | 78.7 ± 10.4 | 82.0 ± 08.2 | 79.5 ± 06.0 |

NSCLC, indicating progressively more selective and causally aligned attention. Importantly, AUC remains stable within a narrow range, showing that the improvement in interpretability comes at a moderate and controlled cost in predictive performance.

**Attention Stability** — To assess the sensitivity of attention to architectural changes and to counterfactual intervention, we compute slide-wise Spearman correlation between attention scores produced by different models. For each slide, correlations are computed between pairs of attention vectors of different models. We report results over (i) all slides,

(ii) slides containing tumor (positive slides), and (iii) pathology-relevant regions only (positive patches) in positive slides, as summarized in Table 4. Across both cross-architecture comparisons and baseline–CIA pairs, we observe that correlations are consistently higher when restricted to pathology-relevant regions (positive patches) than when computed over all patches. In contrast, lower correlations at the slide level appear to be driven mainly by variability in the ranking of patches outside annotated tumor regions. This indicates that differences between models and between baseline and CIA variants predominantly affect background or non-discriminative regions, while the relative ordering of tumor-associated patches remains more stable. Our approach, when applied to various MIL methods, thus does not degrade the high attention of important patches if the MIL baseline was already correct, but instead corrects the attention of unimportant patches, leading to a more explainable and interpretable attention.

## 7. Conclusion

In this work, we questioned the reliability of attention as an explainability proxy for attention MIL models, and presented CIA-MIL, a causal attention learning framework for MIL that explicitly enforces the causal contribution of attention to model prediction through counterfactual supervision. Through extensive experiments on tumor subtyping, mutation prediction, and metastasis detection benchmarks, we demonstrated that high predictive performance does not necessarily imply reliable interpretability via attention. CIA-MIL consistently improves attention faithfulness while maintaining competitive predictive performance across both out-of-domain and in-domain feature spaces. This study establishes counterfactual causal intervention supervision of attention as a promising mechanism to improve the reliability of attention in MIL, taking a step further towards deploying trustworthy and clinically actionable AI systems in digital pathology.

## Acknowledgments

This work has benefited from state financial aid, managed by the Agence Nationale de Recherche under the investment program integrated into France 2030, project references ANR-21-RHUS-0003, ANR-21-CE45-0007, ANR-23-CE45-0029, ANR-23-IAHU-0002, and ANR-23-IACL-0003 – DATAIA CLUSTER (as part of IA CLUSTER program). Experiments have been conducted using HPC resources from the Mésocentre computing center of CentraleSupélec and École Normale Supérieure Paris-Saclay, supported by CNRS and Région Île-de-France, and resources from GENCI–IDRIS (Grant 2024-AD011015828). The results shown in this paper are part based upon data generated by the TCGA Research Network: https://www.cancer.gov/tcga.

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

## Appendix A. Additional details for the WSI Datasets

Table 5 presents in more detail the description of the different datasets used in our study. Details about the number of samples and classes are included, together with a small description of each of the digital pathology whole slide image data. More details can be identified in the original papers.

Table 5: **Overview of Whole-Slide Image (WSI) datasets used in this study.** For each cohort, we report the data source, clinical or diagnostic task, number of slides, and the distribution of labels.

| Cohort | Description | # Slides | # Labels |
|---|---|---|---|
| **TCGA** (Tomczak et al., 2015) | | | |
| BRCA | WSIs from TCGA-BRCA. Used for histological subtype classification between invasive ductal carcinoma (IDC) and invasive lobular carcinoma (ILC). | 977 | IDC: 779 ILC: 198 |
| NSCLC | WSIs from TCGA-LUAD and TCGA-LUSC. Used for distinguishing between lung adenocarcinoma (LUAD) and lung squamous cell carcinoma (LUSC). | 956 | LUAD: 478 LUSC: 478 |
| LUAD (TP53) | TCGA-LUAD dataset for TP53 mutation prediction directly from H&E WSIs. Labels correspond to mutation status (mutated vs wild-type). | 427 | TP53 WT: 199 TP53 Mut: 228 |
| **Camelyon 16 Challenge** (Bejnordi et al., 2017) | | | |
| Camelyon16 (Train) | Lymph node metastasis detection dataset. Training subset of whole-slide images (H&E) annotated for the presence of tumor metastasis. | 270 | Normal: 159 Tumor: 111 |
| Camelyon16 (Test) | Official test subset from the Camelyon16 challenge. | 129 | Normal: 80 Tumor: 49 |

## Appendix B. Implementation Details about Attention MIL

We provide additional details here on the attention mechanism used in Attention-Based Multiple Instance Learning (Ilse et al., 2018) used in our work. A bag $B = \{x_i\}_{i=1}^{N}$ consists of $N$ instances, each of which is first transformed into a low-dimensional embedding:

$$\mathbf{z}_i = f(x_i), \qquad i = 1, \dots, N. \tag{7}$$

These embeddings are aggregated using an attention operator to obtain a bag-level representation:

$$\hat{\mathbf{Z}} = \sum_{i=1}^{N} a_i \mathbf{z}_i. \tag{8}$$

The bag prediction is then obtained by applying a classifier to the aggregated representation:

$$\hat{Y} = \varphi(\hat{\mathbf{Z}}). \tag{9}$$

We adopt a gated attention (GA) mechanism (Dauphin et al., 2017) to produce more expressive attention scores. The unnormalized attention score for instance $i$ is computed as:

$$u_i = \mathbf{w}^\top \left( \tanh(\mathbf{V}_1 \mathbf{z}_i) \odot \sigma(\mathbf{V}_2 \mathbf{z}_i) \right), \tag{10}$$

where

- $\mathbf{V}_1, \mathbf{V}_2 \in \mathbb{R}^{L*M}$ are learnable projection matrices,

- $\mathbf{w} \in \mathbb{R}^L$ is a learnable attention vector,

- $\odot$ denotes element-wise multiplication,

- $\sigma(\cdot)$ denotes the sigmoid nonlinearity.

The normalized attention value is obtained using a softmax over all instances:

$$a_i = \frac{\exp(u_i)}{\sum_{j=1}^{N} \exp(u_j)}. \tag{11}$$

Substituting Eq. 11 into Eq. 8, we obtain the gated attention aggregation:

$$\hat{\mathbf{Z}} = \sum_{i=1}^{N} \frac{\exp(u_i)}{\sum_{j=1}^{N} \exp(u_j)} \mathbf{z}_i. \tag{12}$$

## Appendix C. Perturbation Analysis

To ensure a fair and meaningful assessment of attention faithfulness, all perturbation-based analyses are performed **only on correctly classified slides**. This choice follows standard practice in explainability evaluation, as perturbation curves computed on incorrect predictions may reflect model failure rather than the quality of the explanation. Restricting the analysis to correctly predicted bags ensures that attention faithfulness is evaluated conditional on correct model reasoning. Unless stated otherwise, patches are progressively removed according to Eq. 6, and the model output $s(k)$ is recorded at each perturbation step $k$. Empty bags resulting from complete removal are represented by zero-valued inputs.

For quantitative comparison across models and tasks (Tables 1, 2, 3 and Fig. 4), we report AUPC as a relative area, normalized with respect to the unperturbed prediction.

This normalization accounts for differences in baseline confidence across models and tasks, enabling more meaningful cross-model comparisons.

For visualization purposes (box plots of perturbation curves on 2), we report the raw, unnormalized perturbation trajectories $s(k)$ without dividing by $s(0)$. This choice allows a direct and interpretable comparison of how predictive confidence degrades as a function of the perturbation level, without rescaling the curves. Importantly, this visualization choice does not affect the relative ordering of methods, but provides a more intuitive depiction of the perturbation curves in the overall figure.

