# OpenReview forum: "Counterfactual Intervention in Attention Multiple Instance Learning For Digital Pathology"
_MIDL.io/2026/Conference — MIDL 2026 Poster_

### Official Review · Reviewer_G9ci · 2026-01-06

**Confidence:** 4
**Preliminary Rating:** 2
**Final Rating:** 4

**Summary:**

The paper proposes Counterfactual Intervention for Attention Learning (CIA-MIL) by adding an auxiliary classification loss that leverages the difference between predictions from learned attention and counterfactual predictions computed using non-informative (random or uniform) attention, encouraging attention to play a causally meaningful role in the final decision. Attention faithfulness is assessed with Attention-Based Perturbation Analysis, which quantifies instance importance by measuring how rapidly prediction confidence degrades when top-attended instances are removed. To demonstrate generalization, the method is evaluated across multiple TCGA cohorts and with two feature extractors (ResNet50 and UNI), showing that it preserves strong classification performance while consistently improving attention-faithfulness metrics across datasets and backbones.

**Strengths:**

1. Computationally efficient. The method performs interventions at the attention-pooling level rather than at the image/instance level, so it only requires one additional forward pass to obtain the counterfactual prediction.
2. Well-structured and clearly written. The paper is organized logically, making the motivation, method, and evaluation easy to follow.
3. Strong empirical coverage. The authors compare against relevant baselines and evaluate on multiple benchmarks, which helps contextualize the gains.
3. Model-agnostic and lightweight integration. The approach can be applied without increasing model capacity or changing the architecture, making it broadly compatible with existing attention-based MIL pipelines.

**Weaknesses:**

1. Unclear generalization of the proposed benefits. While the method improves attention faithfulness, it is not fully clear how consistently these gains translate across settings. For example, on Camelyon16, Attention-Challenging MIL achieves very strong performance, which may be largely attributable to its multi-branch design rather than the causal objective considered here.

2. Mixed evidence from ablations. In the ablation study, there are cases where disabling the causal term (i.e., setting $ \lambda =0$) yields higher predictive performance, raising questions about when the causal regularization is beneficial and how sensitive the method is to $\lambda$.

3. Missing comparisons to closely related causal MIL methods. The paper does not compare against prior interventional/deconfounding MIL approaches such as IMIL (Interventional MIL with Deconfounded Instance-Level Prediction) and IBMIL (Interventional Bag MIL). Including these baselines would strengthen the claim of robustness and generalizability and clarify how CIA-MIL differs empirically from existing causal MIL frameworks.

**Detailed Comments:**

One practical strength of this work is that it is model-agnostic and can be integrated as an auxiliary loss with minimal architectural changes. It would be helpful to further emphasize this point and encourage broader validation, e.g., testing the auxiliary loss on additional attention-based MIL variants or different feature extractors/datasets to better demonstrate its plug-and-play applicability.

**Justification Of Final Rating:**

After considering the authors’ rebuttal and the revised manuscript, I find that the key concerns from my initial review have been largely addressed. The revised paper now clearly positions CIA-MIL as a plug-and-play, model-agnostic framework that primarily targets improved faithfulness and interpretability of inherent attention in MIL rather than systematic gains in predictive performance. The authors also strengthened the discussion of related causal/interventional MIL methods by clarifying that IMIL/IBMIL are deconfounding-oriented approaches aimed at bias removal and performance under confounding, whereas CIA-MIL treats attention as a causal mediator and intervenes on it to yield explanations that more directly account for predictions. Crucially, this distinction is now empirically supported by adding IBMIL as a reference baseline and providing a comparative table separating performance improvements from explainability/attention-faithfulness gains, making the complementary relationship between the approaches more convincing and the overall technical claims better justified.

**Justification Of The Preliminary Rating:**

I have a solid understanding of the proposed method and its motivation, and I find the idea of enforcing counterfactual attention effects via an auxiliary loss to be clearly presented and easy to follow. However, after reviewing the related literature, I believe there is already a substantial body of prior work on causal/interventional MIL, and the paper does not include direct empirical comparisons to several closely related approaches. This omission makes it difficult to assess the incremental contribution and the generalizability of the proposed framework relative to existing causal MIL baselines, which influenced my preliminary rating.

**Questions To Address In The Rebuttal:**

1. What is the main contribution of the paper? The proposed counterfactual attention intervention framework (CIA-MIL) does not appear to primarily aim for higher predictive performance, so the authors should clearly articulate the core novelty and the intended impact (e.g., improved attention faithfulness/causal alignment rather than AUROC gains).

2. Can the authors provide deeper evidence that the causal objective improves attention quality? Beyond AUPC, it would be useful to more directly inspect or quantify how the attention distributions change when the causal term is enabled (e.g., qualitative visualizations, stability/consistency analyses, or localization-focused metrics).

3. Why were IMIL and IBMIL excluded from benchmarking? Since IMIL (Interventional MIL with Deconfounded Instance-Level Prediction) and IBMIL (Interventional Bag MIL) are closely related causal/interventional MIL approaches, the authors should justify their exclusion and/or clarify whether CIA-MIL is complementary or expected to outperform them under comparable settings.

---

> ### Author Response · Authors · 2026-01-25
>
> We thank the reviewer for their detailed and constructive feedback. We address their concerns point by point below.
>
> **What is the main contribution of the paper?**
>
> The main contribution of this paper is the presentation of a novel, modular and lightweight **counterfactual attention intervention framework (CIA-MIL)** that explicitly treats attention as a causal mediator and intervenes on it during training. To our knowledge, this is the first work to explicitly intervene on attention itself in MIL for the purpose of improving explainability.
> We agree that improvements in attention faithfulness do not necessarily translate into gains in downstream performance, and we do not claim otherwise. To better assess generalization and plug-and-play applicability, we have now integrated CIA into multiple MIL backbones (ABMIL, CLAM, ACMIL) and evaluated it across heterogeneous tasks and feature extractors.
> To provide a compact overview without relying on individual datasets, we report the mean AUC and mean AUPC averaged across all TCGA task–feature combinations for representative models:
>
> | Model      | Avg. AUC ↑ | Avg. AUPC ↓ |
> |------------|------------|-------------|
> | ABMIL      | 86.28      | 76.18 |
> | Ours       | **86.55**  | **68.87** |
> | CLAM       | 85.90      | 75.28 |
> | CIA-CLAM   | 85.50      | 70.00 |
> | ACMIL      | 86.28      | 75.92 |
> | CIA-ACMIL  | 86.20      | 71.42 |
> All these extra experiments are included in Table 1 in the updated paper.  Also, as suggested by the reviewer, we updated the contributions of our paper, by adding a comment at the end of our introduction to explicitly state that “we do not claim to provide a method to increase downstream performance, but rather to maintain the performance of current MIL methods while improving their explainability, as this is crucial when considering the safe deployment of medical imaging techniques.”
>
> **Can the authors provide deeper evidence that the causal objective improves attention quality?**
>
> We revised the formulation of AUPC to compute it  only on correctly classified bags, consistent with prior work on explainability assessment [Hense et al., 2024,Early et al., 2023]. This ensures that attention faithfulness is evaluated conditional on correct model reasoning, rather than penalizing explanations of incorrect predictions. This revision (updated Tables 1, 2, 3 and Figure 2, 4)  also enables more reliable conclusions for challenging tasks such as LUAD-TP53 mutation prediction, where predictive performance is lower.  This revision leads to more insightful and consistent trends, particularly for challenging tasks such as LUAD-TP53, where predictive performance is lower, while for the rest of the tasks changes were very minimal. More details about the way that the attention-based perturbation analysis is conducted are provided in Appendix.C.
> To move beyond reliance on AUPC alone, we expanded the evaluation with complementary instance-level metrics on Camelyon16, where patch-level annotations are available:
> - **AOPCR**, contrasting targeted vs. random perturbations to assess whether attention rankings meaningfully guide the model’s decision;
> - **AUPRC**, treating attention as instance-level predictions;
> - **PG@5**,  measuring whether at least one of the top 5 ranked patches overlaps a metastatic.
>
>  All these extra evaluations are included in Table 3. Overall, we see that CIA-MIL is not shifting attention away from pathology-relevant regions, but is improving its self-explanatory inherent attention. Not all labeled positive patches are of the same degree of positiveness or tumor gravity, therefore, CIA can help better discriminate and rank patches with respect to their effect on the model’s outcome.
> In addition, we performed an analysis of the correlation between slides attentions as reported in table 4 in the updated paper. These results show that CIA does not substantially alter which regions are attended: correlations remain consistently higher on pathology-relevant (positive) patches than on background regions. Differences in attention are primarily driven by non-tumor patches, indicating that CIA refines the ranking and prioritization of attention rather than redistributing it away from relevant regions: CIA, when applied to various MIL methods, thus does not degrade the high attention of important patches if the MIL baseline was already correct, but instead corrects the attention of unimportant patches, leading to a more explainable and interpretable attention.  This provides deeper evidence that the causal objective improves attention quality in a controlled and grounded manner.

---

> > ### Author Response · Authors · 2026-01-25
> >
> > **Ablations and sensitivity to the causal term.**
> >
> > We acknowledge that disabling the causal term can occasionally yield slightly higher predictive performance. Our results suggest that the effect of the intervention is task-dependent: when bag-level performance is already near saturation, like in CAMELYON16, CIA primarily affects attention faithfulness; in lower-signal settings like LUAD-TP53, it can also stabilize training as discussed in ‘TCGA Benchmark’ in Section 5.
> >
> > **Relation to IMIL and IBMIL.**
> >
> > Although our work leverages causal reasoning, CIA-MIL is not designed for the same purpose as prior interventional or deconfounding MIL approaches such as IMIL and IBMIL. These methods primarily aim to remove bias introduced by confounding variables (e.g., staining or acquisition artifacts). In contrast, our use of causality is *methodological rather than corrective*: CIA-MIL explicitly treats attention as a causal mediator and intervenes on it to control its effect on the downstream prediction. The objective is not bias removal, but improving the **faithfulness** of inherent attention mechanisms: if multiple attention distributions can lead to the same predictions, the goal of CIA is to select an attention that directly explains the predictions without the need to post-hoc recalculation of explainability measures, which can introduce additional computational costs.
> > For this reason, CIA-MIL is not expected to systematically outperform IMIL or IBMIL in terms of predictive performance, nor is it intended to replace deconfounding-based frameworks. Instead, the approaches addressed are conceptually complementary. To empirically ground this discussion, we now include IBMIL as a reference baseline on Camelyon16 (Table 3). While IBMIL improves performance via deconfounding, our results show that explicitly controlling attention as a mediator yields complementary improvements in attention faithfulness, supporting the distinction between the two approaches. More importantly, from Table 4, applying IBMIL on top of ABMIL does not change the attention distribution across slides, IBMIL and ABMIL attentions exhibit high correlation on the test slides.

---

### Official Review · Reviewer_jfQh · 2026-01-10

**Confidence:** 3
**Preliminary Rating:** 3
**Final Rating:** 4

**Summary:**

This paper investigates attention mechanisms in digital pathology. The authors investigate the reliability of attention-based multiple-instance learning through a perturbation analysis whereby the effect on prediction is observed when the most attended patches are removed form the input. Based on this analysis, they proposed a "counterfactual intervention" variant of MIL wherein an additional loss function enforces a prediction difference between the prediction with the model's attention distribution, and the prediction with a random or uniform distribution. There are evaluations on multiple datasets.

**Strengths:**

The analysis presented in this paper is interesting and important, because attention-based MIL is often presented as interpretable. The experiments are thorough and cover a range of datasets. The proposed additional loss is simple and well-motivated

**Weaknesses:**

It occurs to me that minimising the AUPC only makes sense if the number of positive patches/instances is a very small fraction of the total slide. This may make sense in some application (e.g. metastasis detection) but not so much in others (such as histological subtyping).

Overall, the results seem fairly inconclusive. The proposed method does not lead to a lower AUPC in most cases, and there are not obvious trends in the trade-off between performance and the AUPC. The paper tends to overstate its findings in some places.

**Detailed Comments:**

No further comments

**Justification Of Final Rating:**

The paper is improved after a round of revisions, and the additional metrics strengthen the case for the suitability of the method. The analysis is thorough and addresses a very pertinent topic that is very relevant to the MIDL community

**Justification Of The Preliminary Rating:**

The analysis thorough and addresses a very pertinent topic that is very relevant to the MIDL community, but the results are inconclusive and the proposed loss does not consistently lead to improved attention distributions.

**Questions To Address In The Rebuttal:**

Please address the suitability of reducing AUPC across different classification tasks

---

> ### Author Response · Authors · 2026-01-25
>
> **Suitability of AUPC**
>
> We thank the reviewer for the insightful comments regarding the interpretation of AUPC. We first clarify that AUPC is not used as a training objective in our work, nor is the model optimized to minimize it. Instead, AUPC is reported **post hoc** as an explainability metric, following prior MIL explainability studies, to assess how attention explains the model’s prediction [Hense et al., 2024, Early et al., 2023].
>
> We agree that minimizing AUPC is most intuitive in settings where instance-level positives are sparse and well-defined, such as metastasis detection. However, many of the tasks considered in this work like histological subtyping (BRCA, NSCLC) and mutation prediction (LUAD–TP53) do not admit a clear definition of a positive patch. The discriminative signal in these tasks is often diffuse, contextual, or compositional rather than localized. Similarly, in tasks such as survival prediction, patch-level labels are inherently ill-defined: no individual patch can be associated with a survival outcome, and prognostic information emerges from global tissue patterns. In such cases, AUPC should not be interpreted as a localization metric, but rather as a relative indicator of attention faithfulness and thus as an explainability proxy for the model’s prediction. If we suppose that multiple attention distributions can lead to the same predictions, the goal of CIA is to select an attention that directly explains the predictions without the need for post-hoc recalculation of explainability measures.
>
> To ensure a fair and meaningful use of AUPC across tasks, we revised its computation (following also the literature [Hense et al., 2024]) to include only correctly classified bags, such that attention faithfulness is evaluated conditional on correct model reasoning. This revision leads to more insightful and consistent trends, particularly for challenging tasks such as LUAD–TP53, where predictive performance is lower, while for the rest of the tasks, changes were very minimal. These changes are highlighted in Section 5, Tables 1, 2, 3 and Figures 2, 4. More details about the way that the attention-based perturbation analysis is provided in Appendix C.
>
> ***Summary across TCGA tasks and feature extractors (average over 6 settings).***
> To provide a compact overview without relying on individual datasets, we report the mean AUC and mean AUPC averaged across all TCGA task–feature combinations for representative models:
>
> | Model      | Avg. AUC ↑ | Avg. AUPC ↓ |
> |------------|------------|-------------|
> | ABMIL      | 86.28      | 76.18 |
> | CIA-ABMIL       | **86.55**  | **68.87** |
> | CLAM       | 85.90      | 75.28 |
> | CIA-CLAM   | 85.50      | 70.00 |
> | ACMIL      | 86.28      | 75.92 |
> | CIA-ACMIL  | 86.20      | 71.42 |
>
> These averages are reported for summary purposes only. As noted above, AUPC is not optimized during training and is used solely as a post-hoc explainability metric. CIA-augmented models maintain comparable average AUC while exhibiting lower average AUPC relative to their non-intervened counterparts. We emphasize that we do not claim that CIA-augmented ABMIL outperforms domain-specific state-of-the-art MIL architectures. Rather, CIA is a lightweight regularization mechanism designed to improve the faithfulness of inherent attention. To make this point clearer, we updated the contribution in Section 1. Moreover, to highlight the impact of CIA, we have integrated CIA into multiple MIL backbones (CLAM, ACMIL), providing a broader and more balanced view of the performance–explainability trade-off. These updates are summarized in Tables 1 and 2.
>
> Additionally, our new analysis of the attention correlation (Table 4 in the updated paper) suggests agreement between baseline and CIA attention remains consistently higher on pathology-relevant (positive) patches than on background regions. This suggests that the intervention primarily refines the relative ranking of non-informative patches, rather than redistributing attention away from clinically relevant regions. As a result, reductions in AUPC reflect improved attention faithfulness rather than changes in the set of patches driving the prediction.
>
> **Inconclusive results, overstatement of findings**
>
> We respectfully disagree with the reviewer. Our new experiments, summarised in Table 2 on the Camelyon16 dataset (on which dense annotations are available) and include our proposed loss in a wide variety of SOTA MIL methods, highlight that our design consistently leads to better explainability and interpretability. Moreover, the overall performances summarized in the previous comment highlight the consistently better overall performance of CIA on AUPC on TCGA datasets, while maintaining comparable average AUC. To address also the comment of the reviewer, we updated the contributions in Section 1. We hope that now our conclusions and findings are clear and we will be very happy to provide more clarifications during the discussion period.

---

### Official Review · Reviewer_HGgZ · 2026-01-10

**Confidence:** 3
**Preliminary Rating:** 4
**Final Rating:** 4

**Summary:**

The paper addresses the trustworthy issues of using attention maps as interpretability tools in traditional MIL models. And propose a counterfactual attention intervention framework, CIA-MIL. This framework treats the MIL model as a structural causal model and introduces a counterfactual intervention during training to guide the attention more causally aligned with the model’s predictions.  The authors evaluate on multiple WSI datasets using two feature extractors. Experimental results show that the proposed method improves the faithfulness of learned attentions while maintaining competitive classification performance.

**Strengths:**

This paper addresses a practically important problem of traditional MIL models that using attention scores as explanations may not be faithful to the model’s prediction.

The approach proposed by this paper, which encourages the causal relationship between attention and prediction through counterfactual intervention is promising.

The authors perform a comprehensive experiment by comparing with several standard baseline MIL models on multiple datasets.

The results show that the proposed CIA-MIL achieved better faithfulness and also maintained task performance.

**Weaknesses:**

The proposed counterfactual attention intervention could potentially introduce unintended behavior—for example, it may encourage the model to assign high attention to patches that are not truly relevant, so high-attention regions may not necessarily correspond to pathology-critical areas. Beyond the AUPC metric, I suggest validating whether the highest-attention patches are actually class-relevant using an additional localization-style evaluation, such as the Pointing Game metric (e.g., as used in Top-down Neural Attention by Excitation Backprop) on Camelyon 16 dataset. Without such verification, the trustworthiness of the learned attention maps may remain uncertain.

**Detailed Comments:**

In Section 3.3 (Attention-Based Perturbation Analysis), when you remove a subset of patches, how is this implemented in practice? Are the corresponding attention scores set to zero, or are they reassigned to randomized/uniform values similar to the counterfactual attention intervention?

**Justification Of Final Rating:**

In the rebuttal, the authors have provided additional experimental results and clarifications on my initial concerns and questions. After the revision, I consider this a good paper and recommend it for acceptance.

**Justification Of The Preliminary Rating:**

This paper tackles an important problem with a promising approach. The experimental results provide solid support and highlight the potential of the proposed method. Adding disease localization performance would further strengthen the work.

**Questions To Address In The Rebuttal:**

Please consider adding additional evaluations on Camelyon16 to measure how well the highest-attention patches localize tumor-relevant regions, as discussed in the weaknesses above.

---

> ### Author Response · Authors · 2026-01-25
>
> **Additional metrics for localization**
>
> We thank the reviewer for their comment. Following their comment, we extended our evaluation by adding the **AOPCR** metric on Camelyon16 (following [Early et al., 2023]), which measures how informative the attention ranking is compared to random perturbations, thereby explicitly assessing whether attention meaningfully guides the model’s decision. Moreover, we revised the formulation of AUPC to compute it **only on correctly classified bags**, consistent with prior work on explainability assessment [Hense et al., 2024,Early et al., 2023]. This ensures that attention faithfulness is evaluated conditional on correct model reasoning, rather than penalizing explanations of incorrect predictions. This revision also enables more reliable conclusions for challenging tasks such as LUAD-TP53 mutation prediction, where predictive performance is lower.  In addition to **AUPRC**, which treats attention scores as patch-level predictions, we introduce pointing game of top 5 patches **PG@5**, defined as the percentage of metastatic test slides for which at least one of the top-5 most attended patches overlaps an annotated metastasis region. This metric is really focusing on the evaluation of the localization of the provided attention. Overall, our extensive analysis highlights the advantages of CIA-MIL over other baselines on all these metrics, maintaining the performance of current MIL methods while improving their explainability, as this is crucial when considering the safe deployment of medical imaging techniques. All these results are highlighted in Tables 1, 2 and 3 as well as Figure 2. Moreover, these changes are also highlighted in the main manuscript and in particular in Sections 5 and 6.
>
> **Unintended behaviour of the CIA module**
>
> To further assess robustness, we integrated the CIA module into additional MIL models. Results indicate that CIA does not fundamentally alter which regions are attended, but rather improves the **ranking and prioritization** of pathology-relevant regions. Localization performance remains comparable, while explainability metrics improve. From Table 3, we observe that CIA-augmented variants reduce AUPC and improve AOPCR while maintaining comparable AUPRC and PG@5, indicating improved ranking faithfulness without degrading localization. To further contextualize these findings, Table 4 in the updated paper summarizes Spearman correlations across slides between attention pairs of different models. Baseline–CIA pairs exhibit lower global correlation but consistently higher agreement on positive patches, suggesting that CIA, when applied to various MIL methods, thus does not degrade the high attention of important patches if the MIL baseline was already correct, but instead corrects the attention of unimportant patches, leading to a more explainable and interpretable attention. All these are highlighted in Tables 2 and 4, as well as Section 6. We will be happy to provide more clarifications if needed during the discussion period.
>
>
> **How is the Attention-based perturbation analysis implemented?**
>
> AUPC is computed by progressively removing patches in order from the most-attended to the least-attended ones. Perturbations result in zero-vector representations for empty bags. Evaluation is restricted to correctly classified samples to ensure a fair and meaningful assessment of explanation quality. Results on Tables 1,2,3 have been updated. These had been integrated into the main manuscript in Sections 5 and 6. This revision leads to more interpretable and consistent trends, particularly for challenging tasks such as LUAD–TP53, where predictive performance is lower, while for the rest of the tasks, changes were very minimal. More details about the way that the attention-based perturbation analysis is provided in Appendix C.

---

### Author Rebuttal · Authors · 2026-01-25

**Rebuttal:**

We thank the reviewers for their thoughtful, thorough, and constructive feedback.  We appreciate the overall positive assessment of our work, including (i) interesting and important analysis in the paper [HGgZ, jfQh], (ii) the comprehensive experiments and comparisons with other baselines [HGgZ, jfQh, G9ci], (iii) well-motivated, simple and lightweight design of our method [jfQh, G9ci] and the computationally efficient design of our method [G9ci]. Below, we provide point-by-point responses to each reviewer’s comments and indicate the corresponding modifications made to the manuscript. Our revision includes: (i) additional experiments adding our module to more MIL methods, proving better interpretability, (ii) inclusion of extra metrics, (iii) comparisons with other causal intervention methods, and (iv) extensive changes to the manuscript to include the comments of the reviewers. All changes in the revised PDF are highlighted in red to facilitate the review process. We believe that these revisions have strengthened the paper’s contributions, improved clarity and reproducibility, and adequately addressed all reviewer concerns. We hope that with these extra comparisons and evaluation of the impact of our novel, modular and lightweight CIA-MIL is better highlighted. To our knowledge, this is the first work to explicitly intervene on attention itself in MIL to improve explainability.

**Supporting Material:**

/attachment/010b6d7841ca689dc7c1843038809e868b1f6e36.pdf

---

### Comment · Area_Chair_6CHx · 2026-01-28
**Engage in discussion & final rating reminder**

I would like to encourage reviewers to engage with the authors during the discussion phase to clarify any missing or contradictory points.

Please ensure that the final rating is updated by February 1st 2026 (23:59 AoE).

---

### Meta-Review · Area_Chair_6CHx · 2026-02-05

**Recommendation:** Accept (Poster)
**Confidence:** 5

**Metareview:**

All reviewers evaluated the final version of the manuscript positively (three *Weak accept*).

They agree that the major concerns were addressed and that the addition of metrics, along with the clarification of the paper’s position on improving faithfulness and interpretability, constitutes a relevant contribution to the MIDL community.

---

### Decision · Program_Chairs · 2026-02-13

Accept (Poster)